# Osteonal Damage Patterns from Ballistic and Blunt Force Trauma in Human Long Bones

**DOI:** 10.3390/life14020220

**Published:** 2024-02-03

**Authors:** Keira Sexton, Nathalie Schwab, Ignasi Galtés, Anna Casas, Nuria Armentano, Pedro Brillas, Xavier Garrido, Xavier Jordana

**Affiliations:** 1Institute for Interdisciplinary Studies, Faculty of Science, University of Amsterdam, 1098 XH Amsterdam, The Netherlands; keira.sexton@gmail.com; 2Forensic Anthropology Unit, Catalonian Institute of Legal Medicine and Forensic Science (IMLCFC), 08075 Barcelona, Catalonia, Spain; nathaliecorinne.schwab@autonoma.cat; 3Biological Anthropology Unit, Department of Animal Biology, Plant Biology and Ecology, Faculty of Biosciences, Universitat Autònoma de Barcelona (UAB), 08193 Bellaterra, Catalonia, Spain; anna.casast@autonoma.cat (A.C.); nuria.armentano@uab.cat (N.A.); 4Research Group of Biological Anthropology (GREAB), Biological Anthropology Unit, BABVE Department, Universitat Autònoma de Barcelona (UAB), Cerdanyola del Vallès, 08193 Bellaterra, Catalonia, Spain; 5Legal Medicine Unit, Department of Psychiatry and Legal Medicine, Universitat Autònoma de Barcelona (UAB), Cerdanyola del Vallès, 08193 Bellaterra, Catalonia, Spain; 6Donor Center Barcelona Tissue Bank (BTB), Hospital Clínic de Barcelona, 08036 Barcelona, Catalonia, Spain; pbrillas@clinic.cat; 7Mossos d’Esquadra, Unitat Central de Balística i Traces Instrumentals, 08206 Sabadell, Catalonia, Spain; xavi.garrido@gencat.cat; 8Tissue Repair and Regeneration Laboratory (TR2Lab), Institut de Recerca i Innovació en Ciències de la Vida i de la Salut a la Catalunya Central (IrisCC), 08500 Vic, Catalonia, Spain

**Keywords:** forensic anthropology, hard tissue biomechanics, trauma mechanisms, long bone fracture, gunshot trauma, blunt force trauma, bone histology, microcrack

## Abstract

Forensic anthropologists play a key role in skeletal trauma analysis and commonly use macroscopic features to distinguish between trauma types. However, this approach can be challenging, particularly in cases of highly comminuted or incompletely recovered fractures. Histological analysis of microscopic fracture characteristics in fractured bones may thus help provide additional information on trauma type and bone fracture biomechanics in general. This study analysed the extent of microcrack damage to osteons in long bones with blunt force trauma (BFT) and gunshot trauma (GST), from both traumatic death cases and post-mortem experimental fractures. We identified four types of osteonal damage (OD). In traumatic death cases, OD affecting the inside of the osteon and compromising the Haversian canal (type 1) was found to be indicative of BFT. Moreover, OD affecting the cement line (type 3) and interstitial lamellae (type 4) was more common in the GST samples. OD affecting the inside of the osteon without compromising the Haversian canal (type 2) was not found to be indicative of either trauma type. In cases of experimental fractures, our study revealed that post-mortem fractures in dry bone samples featured the highest amount of OD, particularly of type 4. This study also found that the experimentally produced GST featured similar OD patterns to GST death cases. These findings support our hypothesis that there are distinct osteonal damage patterns in human long bones with BFT and GST, which are of relevant value for trauma analysis in forensic anthropology.

## 1. Introduction

Forensic anthropologists contribute to medical–legal investigations into deaths by evaluating mechanisms of trauma in skeletal remains such as blunt, sharp, thermal or gunshot trauma [1]. Forensic anthropological cases include examining not only bones but also bodies which have succumb to advanced taphonomical changes, for example severe decomposition, burning, saponification, scavenger activity or mummification [2]. In cases impacted by these taphonomical factors, traditional reliance on soft tissue to evaluate the mechanical forces that were applied is often no longer possible [3]. Bones, however, are more resistant and reliable in cases impacted by such taphonomical factors and may present an important permanent record of trauma evidence in bones that are affected. Trauma analysis necessitates a comprehensive understanding of injury mechanisms, bone composition and bone biomechanics (influenced by both the bone’s macro- and microstructure) in response to trauma. This knowledge can sometimes point forensic anthropologists towards the sequence of trauma, the type of weapon used, the manner, and cause of death based on the assumption that specific fracture patterns result from different mechanical impacts [4]. Despite the biomechanical differences between gunshot trauma (GST) and blunt force trauma (BFT), it can be challenging to interpret and reconstruct with certainty the type of trauma in fractured long bones [3,5], particularly when dealing with comminuted fractures or incompletely recovered fractured bones.

Bones will resist, begin to fracture, and eventually fail in accordance with the stress–strain curve [6]. In BFT, the ductile bone fails in response to contact with broad or blunt surfaced objects at a low to medium velocity [7]. GST, or ballistic trauma, occurs when high-velocity projectiles penetrate the brittle bone and produce fast-loaded injuries [7]. Such injuries can cause severe bone fragmentation as a result of the instant accumulation and release of massive levels of energy [6]. However, unlike BFT, where elastic deformation is followed by plastic deformation and eventual bone failure; viscoelastic bone in response to the high velocity of bullets acts as a brittle material and fails upon contact [6,8]. The exact biomechanics behind ballistic bone trauma remain controversial as it is difficult to apply the fundamental laws of mechanics to such a transient event [6]. Cranial vault trauma has been well studied compared to long bone injuries, despite these injuries also being key for reconstructing events surrounding a person’s death [3,7,9]. Studies on human long bones are starting to shift from a general description and focus on the fractures’ pathophysiology or the ballistic variables towards a more detailed assessment of macroscopic fracture characteristics themselves [3,5,10].

Although macroscopic analysis is more predominant in terms of bone fracture analysis, microscopic fracture patterns should also be considered as important sources of information [11], particularly for circumstances where only bone fragments are recovered, or the fracture is too comminuted to be reliably assessed macroscopically for trauma type. Histological analysis may also help gain more insight on fracture biomechanics and fracture mechanisms themselves. The principle factor in the bone’s resistance to fracture is its toughness, in other words, the bone’s ability to resist crack propagation at multiple hierarchical levels [6,12]. Despite microcracks (MCKs) diminishing stiffness, they represent a form of defence against crack formation and bone failure through a phenomenon known as microcrack toughening [13]. The mechanics surrounding MCKs are still yet to be fully understood and have mainly been investigated in terms trauma timing in BFT cases [14,15,16]. MCKs, however, are also impacted by trauma circumstances and thus are of forensic relevance [12,13]. As such, it is also important to consider MCKs in the context of trauma typing. Due to the difference in impact energy between BFT and GST, we hypothesised that differences in osteonal damage caused by MCKs would also be observable microscopically.

As highlighted in previous papers, there is a need for further investigation into osteonal microcracking patterns since osteonal MCKs specifically have been associated with fresh fractures [12,15,16]. So far, there has not been a focus on osteonal microcracking patterns in relation to trauma type but only literature on BFT cases and trauma timing [14,15,16]. These studies have demonstrated that in fresh BFT cases, a higher proportion of osteonal MCKs can be observed when compared to dry bone which has greater interstitial MCKs [15,16]. Hence, it is important to further investigate the osteonal microcracking pattern in BFT cases and define the osteonal microcracking pattern in a different trauma type such as GST.

The objective of this study was to explore the possibility of determining the trauma type, BFT or GST, from distinct osteonal damage patterns present in fractured human long bones. This aim was based on the hypothesis that histological analysis could identify microscopic characteristics specific to GST and BFT in the osteons which would enable them to be distinguished. To test this hypothesis, this study evaluated the extent of osteonal damage and the type of osteonal damage observable in human bone samples with different trauma types (GST and BFT). The ability to distinguish between GST and BFT as a result of osteonal damage pattern analysis would be of particular benefit for challenging cases where fractures are comminuted or fragmented.

## 2. Materials and Methods

In this study, we analysed human long bones with blunt force and gunshot trauma resulting from traumatic death and post-mortem experimental cases (Table 1).

### 2.1. Traumatic Death Cases

Five humeri with BFT from medicolegal autopsies were collected as part of complementary medicolegal investigations at the Institute of Legal Medicine and Forensic Science of Catalonia (IMLCFC). Additionally, five bone fragments from four humeri and one femur with GST were provided from skeletal remains recovered from a Spanish Civil War mass grave in Paterna Municipal Cemetery (Valencia, Spain) [17]. Two of the fragments were determined to originate from the bullet’s entry region and the other three fragments were from the exit region.

### 2.2. Experimental Fracture Cases

To assess the effect of taphonomy and histological procedure on osteonal damage, two healthy, unfractured and consensually donated fresh human humeri were dissected from the upper limbs by the Blood and Tissue Bank (Catalan Department of Health) and provided for experimental use after medical rejection for transplantation. The bones (stored at −80 °C) were thawed and the remaining soft tissue up to the periosteum was removed using surgical tools. One of the specimens was used to reproduce GST fracture to compare with traumatic death cases, and the other specimen was used as a control to assess the effect of bone conservation and preparation on osteonal damage production. Additionally, a dry, healthy and unfractured human humerus donated to science was provided by the IMLCFC (collection registered at the Instituto de Salud Carlos III, Reference C.0004241) and used to reproduce post-mortem BFT (taphonomical fracture damage).

### 2.3. Fracture Reproduction

To reproduce BFT in the dry specimen, a three-point bending fracture was produced by a pendulum impact test machine (BFT simulator), which consists of a metal frame and a pendulum with a 5 kg hammer attached. The bone was positioned horizontally (anterior side facing the hammer) and secured to two metal holders with cable ties. Soft rubber was attached to the hammer to prevent direct contact with the bone when hitting the shaft perpendicularly. The anterior bone side experiences compression loading whilst the opposite posterior side breaks under tension loading [15,18].

To reproduce GST, the fresh specimen was experimentally shot at the ballistic laboratory of the Catalonian police, Mossos d’Esquadra. The bone’s diaphysis was placed in a metal mould to which liquid ballistic gel (Clear Ballistics, Greenville, SC, USA) was added and left to solidify [10]. The bone sample was then stabilised vertically on a platform and placed two metres away from the muzzle of a gunshot trauma simulator. The GST simulator was loaded with the most common handgun cartridge worldwide, a 9 mm Luger full metal-jacket [19]. The middle of the anterior diaphysis was targeted using a laser collimator [10].

### 2.4. Bone Preparation and Histological Analysis

For easier fracture examination, the fresh samples (BFT_autopsy and GST_experimental) were macerated to remove remaining fresh tissue. The bones were boiled in a water detergent solution (one cup of detergent for five litres of water) at 90 to 100 °C for three to five hours and further cleaned to remove persisting tissue such as the periosteum [18].

Two control samples were added to assess the possibility of processes such as freezing and boiling or histological procedure forming osteonal microdamage. Specifically, the unfractured frozen bone (Control), stored at −80 °C for just under a year, was cut in half and the epiphyses were removed. One half of the bone was boiled at under 100 °C for approximately four hours. Whilst the other half was defleshed manually, including the periosteum, without boiling, chemical processes or metal equipment.

All bone samples were subsequently fixed and dehydrated following the protocol by Ebacher et al. [12] and de Boer et al. [20]. The bones were placed in a sequential series of ethanol solutions (70%, 80%, 90%, 100%) every 24 h. Bone samples were then embedded in resin and cut transversely 1 cm below the main fracture line with a Buehler IsoMet saw. Bone fragments from the Spanish Civil War mass grave samples (GST_mass grave) were completely embedded in resin and cut transversely in half. All the bones’ embedded surfaces were polished with carborundum powder, cleaned in an ultrasonic bath and dried in an oven at 30 °C. The resin blocks were then glued to microscope slides using an ultraviolet curing glue (Loctite 358) and once dry, cut into 100 µm thin sections using the PetroThin (Buehler, Lake Bluff, IL, USA) cutting machine. The slides were then placed into an alcohol gradient (70%, 96%, 100%) before being fixed with Histolemon. A drop of the mounting medium DPX (Dibutylphthalate Polystyrene Xylene) was then added to the slides to bond the coverslips until polymerised.

Previously, the possibility of histological techniques contributing significantly to MCK propagation was ruled out [15]. Both control samples, frozen only and frozen-boiled, were not found to have any visible MCKs in the cortical bone under a light microscope.

All bone samples were analysed under a light microscope (Leica DMD 108) at a magnification of 4×. Scaled micrographs of the cortical bone were taken and stitched together to create two-dimensional mosaics of all the cortical surface using the Fiji imaging processing package and the MosaicJ plugin [21,22]. Using ImageJ v.1.53t, the cortical bone was digitally delineated into two regions, compression and tension sides for BFT samples and bullet entry and exit sides for GST samples [23]. An indexed grid with subfield areas of 1 mm^2^ was overlayed onto the mosaics. Random letter and number combinations were generated to select 20 subfields per region for analysis (40 total subfields per mosaic). Randomly selected subfields were discarded and redrawn if they had areas without bone tissue such as those near the endosteal or periosteal edge, if obscured by fungi or bacterial damage, or if the bone slice’s thickness prevented visualisation of the bone tissue.

All mosaics were analysed for osteonal damage (OD) defined as osteons (including the cement line) compromised by MCKs. The extent of OD was determined by the percentage of damaged osteons compared to the total osteons (damaged and healthy) in each 1 mm^2^ subfield. Additionally, four different osteonal damage types were investigated (Figure 1). In instances where multiple MCKs damaged the osteon, the MCK with the greatest length (greatest impact) was considered to determine the OD type sustained.

### 2.5. Statistical Analysis

The mean and standard deviation were calculated from the data obtained. Normality of variables was tested using the Shapiro–Wilk test. For comparative inferential analysis, the Mann–Whitney-U and two-way ANOVA tests were used. The Dwass–Steel–Critchlow–Fligner and Tukey’s post hoc tests were also used where applicable. A BFT_autopsy sample was selected to be re-examined by the same observer a month later for both percentage osteonal damage and osteonal damage type. The percentage osteonal damage was also assessed by a different observer. The intra- and inter-rater reliabilities were evaluated using the Wilcoxon signed-rank test and the intra-class correlation (absolute agreement) test. Statistical analyses were performed using Jamovi 2.3 [24,25]. A significance level of 0.05 was set.

## 3. Results

Table 2 shows the summary statistics of total OD and the OD types in all four sample groups (BFT_autopsy, BFT_dry experimental, GST_mass grave, and GST_experimental).

### 3.1. Osteonal Damage Extent

The total OD differs in each sample group. BFT_dry experimental had the greatest OD followed by the GST_mass grave samples, then the BFT_autopsy samples, and lastly, the GST_experimental sample (Table 2). Total OD in the BFT_dry experimental sample was significantly greater compared to all the other sample groups (Dwass–Steel–Critchlow–Fligner test: *p*-values < 0.001).

In BFT_autopsy and BFT_dry experimental sample groups, OD was greater in the tension side compared to the compression side (Table 3). These differences were only statistically significant for the BFT_dry experimental samples (Mann–Whitney U: *p*-value = 0.026) and not for the BFT_autopsy samples (Mann–Whitney U: *p*-value = 0.213).

In the GST_mass grave sample, OD was greater in the exit side compared to the entry side. Contrarily, in the GST_experimental sample, OD was greater in the entry side. However, for both samples, the differences were not statistically significant (Mann–Whitney U: *p*-value = 0.660 and *p*-value = 0.463, respectively).

### 3.2. Osteonal Damage Type

The distribution of the different OD types in the selected subfields of the four sample groups are represented in Figure 2.

Examples of osteonal damage types observed in 1 mm^2^ of the four sample groups represented in Figure 2 can be seen in Figure 3. When comparing OD types, there were statistically significant differences within and between the different sample groups (ANOVA: *p*-value < 0.001).

Within the BFT_autopsy samples, OD type 1 was significantly greater than the three other OD types (Figure 4) (Tukey’s Post Hoc test: *p*-values < 0.001). Within the BFT_dry experimental sample, OD types 1 and 4 were significantly greater than the other OD types (Tukey’s Post Hoc test: *p*-values < 0.001). Within GST_mass grave and GST_experimental samples, OD types 3 and 4 were the most common when compared to OD types 1 and 2 (Table 2); however, these differences were not statistically significant (Figure 4).

Between the traumatic death cases, OD type 1 was significantly greater in BFT_autopsy sample group when compared to GST_mass grave sample group (Tukey’s test: *p*-value < 0.001). OD types 3 and 4 were significantly greater in GST_mass grave sample group when compared to the BFT_autopsy sample group (Tukey’s test: *p*-value = 0.024 and *p*-value = 0.021, respectively).

Between the BFT samples, OD type 1, 2 and 4 in the BFT_dry experimental sample group are significantly greater compared to BFT_autopsy sample group (Figure 4) (Tukey’s test: *p*-values < 0.001, 0.028 and <0.001, respectively). Between GST samples (mass grave and experimental), there were no statistically significant differences in all four OD types (Figure 4) (Tukey’s test: *p*-values = 1.000, 0.993, 1.000 and 1.000, respectively).

### 3.3. Observer Reliability Tests

There were no significant differences between the first and second data recorded by the same observer for both total osteonal damage (Wilcoxon’s test: *p*-value = 0.909) and osteonal damage type (Wilcoxon’s test: *p*-value = 0.673). The intra-class correlation coefficient (ICC) was high for both total OD and OD type (ICC > 0.9) indicating excellent intra-rater agreement. Similarly, in terms of total osteonal damage there were no significant differences between the data recorded by the first observer and a different observer (Inter-rater reliability: *p*-value = 0.844), with an ICC of 0.798 indicating good inter-rater agreement.

## 4. Discussion

Forensic anthropologists need more tools at their disposal to better enable trauma type assessment and reconstruction of death circumstances, including an improved understanding of GST biomechanics. Distinguishing between BFT and GST in long bones can be challenging when fractures are comminuted or incompletely recovered [3,5,26]. Rather than macroscopic analysis of fracture patterns, this study focused on the histological consequences of trauma, specifically, on the osteonal damage. Through analysing the osteonal damage extent and the types of this damage, we aimed to define a distinct pattern in BFT and GST and evaluate the possibility of utilising histological analysis on fractured human long bones to distinguish between these two trauma types.

### 4.1. Osteonal Damage Extent

It has been well documented that approximately 80 to 90% of MCKs are located in the broad and brittle interstitial area of the cortical bone as it is easier for them to initiate there due to the presence of large hydroxyapatite crystals, reduced osteocyte density, and limited bone remodelling [15,16,27,28]. Nevertheless, Ebacher et al. [12], Winter-Buchwalder et al. [15], and Schwab et al.’s [16] studies have all noted the presence of microcracking patterns related with bone trauma and highlighted the importance of focusing on osteonal MCKs in fresh fracture histological analysis [15]. The results of our study on osteonal microcrack damage, when comparing GST and BFT traumatic death cases, showed that total osteonal damage is greater in GST despite the difference not being significant. This difference could be linked with the lower loading rate in BFT when compared to GST samples. In BFT, microcracks propagate more slowly compared to GST taking the load away from the surrounding bone material [29]. In GST, a stress wave goes through the bone at the speed of sound due to the impulsive loading, which in turn initiates and propagates several MCKs with increased velocity [29]. The crack’s increased velocity results in less energy absorption during its travel and further propagation, resulting in highly comminuted fractures [13]. Another important factor to consider is bone’s viscoelastic nature. In BFT, bone behaves as a ductile material and is elastic and flexible allowing for more energy to be transferred to the stretch mechanism, resulting in less damage compared to the GST samples [6]. In GST, bone behaves as brittle material when fracturing due to the rate of energy transfer being too high for viscoelastic compensation [6]. This difference in behaviour could explain the increased OD in GST. Having said this, the differences are not significant between GST_mass grave and BFT_autopsy samples nor the GST_experimental sample. A potential explanation for this may be impact-related nature of the BFT_autopsy cases, consisting of traffic accidents and falls, which transfers more kinetic energy, involves triaxial stress states and multiple high impact points on the bone, unlike a fracture caused by a relatively simple three-point bend [16,30].

In ballistic trauma, fracture patterns are linked to the energy transferred by the bullet when impacting, penetrating, and perforating the bone and the bullet’s material characteristics, kinetic energy, impact profile, and deformation/fragmentation as well as the biological characteristics of the target tissue (elasticity, density and cohesiveness) [9,10]. In particular, bone fracture patterns are attributable to the magnitude and rate of loading force, which determine the amount and rapidity of stress applied to bone before it deforms or fractures [6]. The results of our study indicate that total OD is greater, although not significantly, in GST_mass grave samples when compared to GST_experimental. It could be hypothesised that GST_mass grave had more OD due to differences in energy transfer between the rifles assumed to be used during the Spanish Civil War and the handgun used in the experimental fracture process [6]. As the projectile’s velocity would be greater in a rifle, so too would be the fracturing and fragmentation of the target bone [9]. This may be potentially contributing to the increased OD, although not significant when compared to GST_experimental.

The difference in OD was also investigated between the compression and tension regions of the BFT samples and the bullet entry and exit regions of the GST samples. BFT_dry experimental was the only sample group to have significantly greater OD in one region, the tension region. BFT_autopsy also had greater OD in the tension region despite not being significant. These findings are in agreement with the fact that in BFT we see bending where bones are said to be stronger in compression, with the tension side being the weakest and failing first [13,30]. Both GST_mass grave and GST_experimental samples had greater OD in differing regions (exit and entry, respectively). The lack of significant differences in terms of OD between regions in GST samples could be explained by it not being possible to observe clear-cut regions of forces at play in GST as it is known for BFT. Long bones hit by a bullet are primarily fractured due to the hydraulic pressure built up in the bone marrow (generated by the temporary cavity), which acts in all directions across the whole cortical bone, causing the diaphysis to split open from within and shatter [5,31].

### 4.2. Osteonal Damage Types

This study has differentiated four types of osteonal damage: damage inside the osteon affecting the Haversian canal (type 1), osteonal damage without compromising the Haversian canal (type 2), damage in the osteon’s cement line only (type 3) and, lastly, damage affecting both the cement line and interstitial lamellae (type 4). Our results showed that OD types, except for OD type 2, do differ significantly between GST and BFT. This is in line with Piekarski’s [32] hypothesis that the velocity of trauma has an effect on how the fracture line propagates in bone. Although our study differed by considering MCKs in terms of osteonal damage and not fracture lines, MCKs in densely damaged areas eventually coalesce to form fracture lines [33,34]. Osteonal damage affecting the inside of the osteon and the Haversian canal (OD type 1) was significantly greater in BFT compared to GST samples. OD type 1 therefore appears to be an indicator of BFT. In the literature, the void structures inside the osteon such as the Haversian canals, osteocyte lacunae and canaliculi are said to act as stress concentration sites where MCKs may initiate [12,13,15,16]. Winter-Buchwalder et al.’s [15] study observed that the osteonal MCKs in BFT ran from the Haversian canals to the cement lines which supports our findings of OD type 1 being the most prevalent in the BFT autopsy samples. Our results differ, however, from conclusions of other studies using animal bones, which suggested a preference for cracks to follow the path of least resistance, the cement line, at low loading rates [32,35]. Caution should be taken when interpreting findings from animal bones as they do not have the same biomechanical properties and fracture production as human bone [5,10,36]. Our results show that depending on the trauma, the type of osteonal damage differs significantly in human bone. At low loading rates, in BFT, osteonal damage affecting the osteon’s Haversian canals was found to be the most paradigmatic.

In the GST_mass grave samples, osteonal damage affecting the cement line (type 3) and the interstitial lamellae (type 4) were significantly greater when compared to the BFT_autopsy samples. This suggests that unlike BFT_autopsy, the osteonal damage in GST_mass grave is not primarily confined to the inside of the osteon alone. The MCKs impact significantly more the interstitial lamellae and cement line in GST_mass grave compared BFT_autopsy.

When considering the GST (mass grave and experimental) samples alone, there were no significant differences in terms of OD types present within the sample groups themselves nor between them, although OD types 3 and 4 were the most common in both samples. This supports literature stating that at higher strain rates MCKs propagate through the microstructure without preference [32]. This differs from our findings in BFT samples were OD type 1 is significantly greater than the three other OD types. Although blast trauma differs from ballistic trauma, Pechníková et al. [35] found fracture lines going through the microstructure indiscriminately in blast trauma (high velocity) as we found with GST [37].

When it comes to differences between GST samples, the OD type pattern present in the traumatic death cases from a mass grave of the Spanish Civil War and the experimentally reproduced GST case are similar. The inclusion of samples from a Spanish Civil War mass grave (PMI of 85 years) meant the impact of taphonomical factors on osteonal damage assessment could be explored. The outcomes indicate that with approximately 80 years of taphonomical exposure, histological analysis of bone fragments for osteonal damage could still be performed. These findings also provide support for the experimental GST methodology used to simulate GST sustained during life. In terms of experimental GST, future studies should consider investigating the use of additional axial compression to better simulate intra vitam conditions, different bullet types, impact velocities and shot distances.

When considering the post-mortem BFT dry sample (PMI of 20 years), BFT_dry experimental had significantly the greatest total OD and the greatest OD types 1 and 4 out of all the sample groups. This aligns with previous findings indicating taphonomical bone breaks result in more interstitial MCKs when compared to fresh fractured bone [15]. However, type 4 OD was also greater in both GST samples when compared to BFT_autopsy, although not comparable to that of the dry specimen. Perhaps, OD type 4 is therefore an indicator of bone behaving as a brittle material as is expected with the rapid loading rate in the GST cases and with the dry nature of BFT_dry experimental, preventing the bone from withstanding as much strain or elastic deformation [6]. Interestingly, BFT_dry experimental also had significant amounts of OD type 1, supporting our hypothesis that type 1 OD is an indicator of BFT. Together, the prevalent OD types 1 and 4 in BFT_dry experimental align with BFT and dry bone characteristics, respectively. Osteonal damage type is therefore not only related to moisture or elasticity of the bone but trauma type as well.

In a forensic context, the histological analysis of human long bones with trauma for osteonal damage types could help to distinguish between BFT and GST. This would be of particular importance for forensic cases where the fracture is very comminuted preventing the interpretation of the mechanisms involved and thus establishing the trauma type from the small fragments available. The prevalence of certain OD types present in the sample could therefore assist the forensic practitioner in drawing conclusions on the trauma type. As found in this study, a prevalence of OD type 1 would be indicative of BFT and a prevalence of OD types 3 and 4 would be characteristic of GST. Increased OD type 4 was also found to be indicative of post-mortem damage in dry bones.

### 4.3. Limitations

As this study uses real human remains, the main limitation is its small sample size. In future, by building on the foundations set in this paper, the analysis of a larger sample size would permit, in terms of potential error rate, enough discriminatory power for the accurate classification of GST and BFT. A larger sample size would also allow for the exploration of important variables such as age and sex. Another limitation to consider in a forensic context is the destructive nature of the histological analysis. However, as evidenced in the Spanish Civil War samples with GST, it is not necessary to cut the whole cortical bone as using only a small fragment from a comminuted fracture was sufficient to record the osteonal damage extent and types. Additionally, other trauma types should also be investigated in human long bones such as sharp force trauma and blast trauma.

## 5. Conclusions

This exploratory study analysed osteonal microdamage in human long bones with blunt force and ballistic trauma and categorised it into four types of osteonal damage. Our study revealed that in trauma death cases, there is more osteonal damage in gunshot trauma compared to blunt force trauma, although the difference is not significant. When considering the osteonal damage types, in BFT, the majority of osteonal damage significantly affected the inside of the osteon compromising the Haversian canal (type 1). However, in GST, osteonal damage impacting the cement line (type 3) and the interstitial lamellae (type 4) were more prevalent. Our study additionally revealed that post-mortem fractures in dry bones are characterised by a higher amount of osteonal damage, particularly the osteonal damage affecting the interstitial lamellae (type 4). Notably, the experimental production of GST was found to yield similar osteonal damage extent and types when compared to traumatic GST death cases which have been exposed to taphonomical factors. Overall, we can conclude that there are distinct osteonal damage patterns in human long bones with GST and BFT, and in fresh peri-mortem trauma and in post-mortem damage. Therefore, the reported osteonal damage differences may be of value when exploring trauma type in forensic anthropology.

## Figures and Tables

**Figure 1 life-14-00220-f001:**
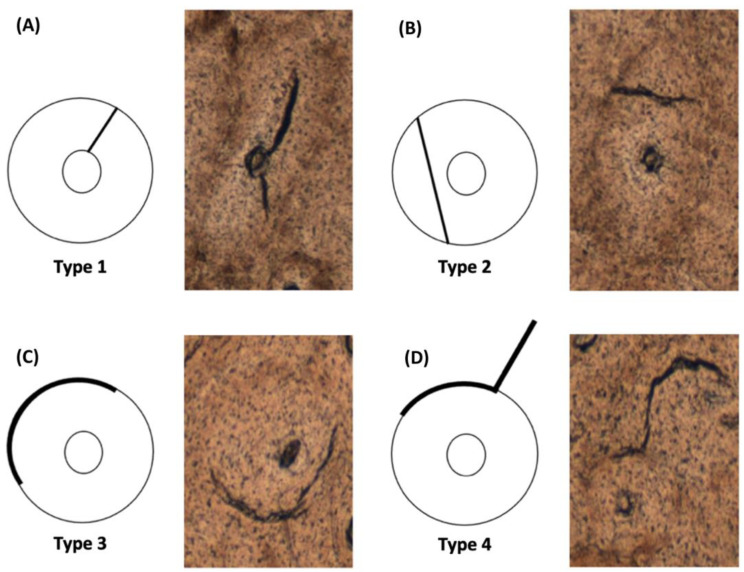
Osteonal damage (OD) types: (**A**) type 1, the microcrack (MCK) is inside the osteon and has contact with the osteon’s Haversian canal, (**B**) type 2, the MCK is inside the osteon but does not have contact with the Haversian canal, (**C**) type 3, the MCK present only in the osteon’s cement line, and (**D**) type 4, the MCK damages the interstitial lamellae and the osteon’s cement line. The circle’s inner ring represents the osteon’s Haversian canal, the outer ring represents the osteon’s cement line, the area in between both rings represent the osteonal bone and the bold line represents a MCK.

**Figure 2 life-14-00220-f002:**
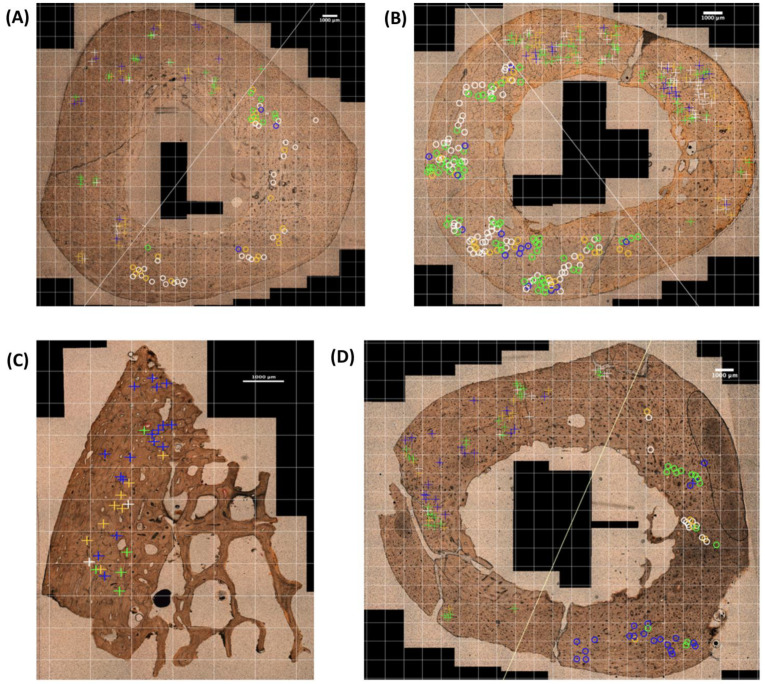
OD types observed in the selected subfields of the cortical bone cross sections. (**A**) BFT_autopsy bone sample, (**B**) BFT_dry experimental bone sample, (**C**) GST_mass grave bone sample, and (**D**) GST_experimental bone sample. The crosses on the bone indicate a damaged osteon (damaged by a MCK) in the compression region of BFT samples and bullet entry region of GST samples. The circles indicate a damaged osteon in the tension region of BFT and bullet exit region for GST. The colour of the crosses and circles indicate the OD type: white = OD type 1, orange = OD type 2, blue = OD type 3, and green = OD type 4.

**Figure 3 life-14-00220-f003:**
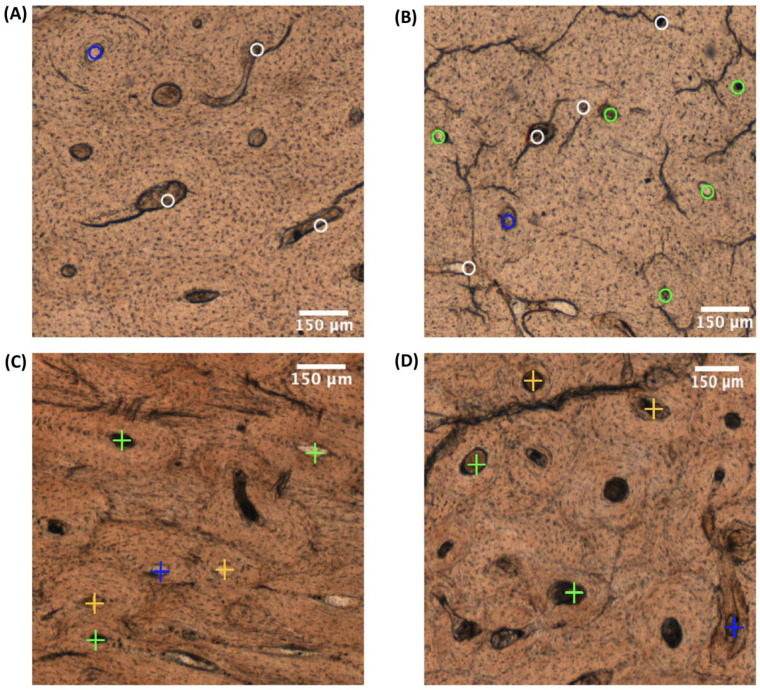
Osteonal damage patterns (1 mm^2^): (**A**) BFT_autopsy case, tension side; (**B**) BFT_dry experimental case, tension side; (**C**) GST_mass grave, entry side; and (**D**) GST_experimental, entry side. The colour of the crosses and circles indicate the OD type: white = OD type 1, orange = OD type 2, blue = OD type 3, and green = OD type 4. The crosses on the bone indicate a damaged osteon (damaged by a MCK) in the bullet entry region of GST samples. The circles indicate a damaged osteon in the tension region of BFT. The colour of the crosses and circles indicate the OD type: white = OD type 1, orange = OD type 2, blue = OD type 3, and green = OD type 4.

**Figure 4 life-14-00220-f004:**
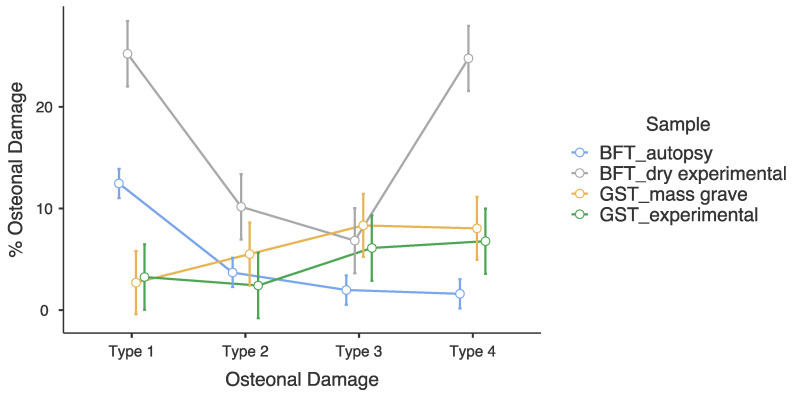
Differences in OD types between sample groups based on estimated marginal means in two-way ANOVA.

**Table 1 life-14-00220-t001:** Study samples’ description.

Sample	Trauma	Sample Name	Origin	PMI *	Sample Size	Bone	Age (yrs)	Sex
Traumatic Death Cases	BFT	BFT_autopsy	Medicolegal autopsies	12–24 h	5	Humeri	36–68	Male
	GST	GST_mass grave	Spanish Civil War mass grave	85 years	5	4 Humeri/1 Femur	30–50	Male
Experimental Fracture Cases	BFT	BFT_dry experimental	Dry bones form IMLCFC collection	20 years	1	Humerus	65–70	Male
	GST	GST_experimental	Fresh bone from cadaver donor	24 h	1	Humerus	65	Male
Control	No trauma	-	Fresh bone from cadaver donor	24 h	1	Humerus	65	Male

* PMI: Post-Mortem Interval.

**Table 2 life-14-00220-t002:** Summary of descriptive statistics for total osteonal damage (OD) and OD types (%) within the sample groups.

Sample	Osteonal Damage Type	*n*	Mean	SD
BFT_autopsy	Total	200	19.7	19.3
	1		12.46	17.19
	2		3.7	7.07
	3		1.98	4.4
	4		1.61	4.26
BFT_dry experimental	Total	40	66.6	21.8
	1		25.22	15.28
	2		10.16	9.78
	3		6.83	10.47
	4		24.76	18.16
GST_mass grave	Total	43	24.2	23.9
	1		2.71	5.43
	2		5.52	8.09
	3		8.33	14.64
	4		8.04	11.31
GST_experimental	Total	40	18.6	18.1
	1		3.26	8.7
	2		2.42	4.44
	3		6.11	8.35
	4		6.78	12.22

n = number of subfields (1 mm^2^) analysed.

**Table 3 life-14-00220-t003:** Summary of descriptive statistics for total OD (%) within sample groups’ sides.

Sample	Side	n	Mean	SD
BFT_autopsy	Compression	100	17.3	17.0
	Tension	100	22.1	21.1
BFT_dry experimental	Compression	20	58.1	24.2
	Tension	20	75.1	15.3
GST_mass grave	Entry	20	21.0	19.6
	Exit	23	27.0	27.3
GST_experimental	Entry	20	21.1	19.9
	Exit	20	16.1	16.3

n = number of subfields (1 mm^2^) analysed.

## Data Availability

The data presented in this study are available upon request from the corresponding authors.

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
