# Peer review of "Osteonal Damage Patterns from Ballistic and Blunt Force Trauma in Human Long Bones"

_life, 2024, doi:10.3390/life14020220_

Round 1

Reviewer 1 Report

Comments and Suggestions for Authors

Overall, with its solid research design and methods, the paper makes a significant contribution to knowledge despite the small sample size, which is to be expected for experiments involving real human remains.

Although some findings are not statistically significant, the systematic approach in the discussion unpacked and synthesized many important issues and concepts. And provides a foundation for future research.

The article is very well written and organized. A few minor editorial/grammatical corrections and suggestions are below.  

Line 51-52: Very minor but should be revised as:

 “…and may present an important permanent record of trauma evidence in bones are affected.”

Line 54-56: what about sequence of trauma?

Line 60: should be “dealing with comminuted”

Line 130: “thawed” is a better choice of words than “defrosted”

Author Response

We would like to thank the reviewers for their valuable comments that enhanced the quality of our manuscript. We hereby outline the proposed changes for the editors' and referees' approval.

Editorial/grammatical corrections and suggestions were all addressed and highlighted in yellow in the text.

Line 51-52: Very minor but should be revised as: “…and may present an important permanent record of trauma evidence in bones are affected.”

  • Updated as suggested.

Line 54-56: what about sequence of trauma?

  • Included as suggested.

Line 60: should be “dealing with comminuted”

  • Updated as suggested.

Line 130: “thawed” is a better choice of words than “defrosted”

  • Updated as suggested.

Reviewer 2 Report

Comments and Suggestions for Authors

The paper presents a study investigating the extent and type of Osteonal Damage (OD) present in samples of bones with projectile and blunt trauma, finding that the extent of OD was not significantly different between the two trauma mechanisms, but that certain types (i.e., locations and patterns) of OD differed between the two mechanisms of trauma.  The study is interesting, well-designed, and should be of interest to forensic practitioners.  I have a couple of suggestions that I believe would improve the paper.

No limitations are discussed in the paper, nor suggestions for further study.  One major limitation that comes to mind is that this type of analysis (which uses bone histology) is destructive in nature, requiring the bone to be cut/damaged in the region of the trauma.  This is problematic in a forensic context, where the evidence is significantly altered by the examination, and then can no longer be examined by another practitioner.  This and other limitations of this approach and this study should be addressed.

Further discussion on how this might be applied and interpreted in a forensic case would also be helpful.  Although significant differences were found in certain OD types between the two different trauma mechanisms, what would a practitioner actually need to observe in order to conclude that the fractures resulted from either projectile or blunt trauma forces?  It is not clear how this approach (examining types of OD) would be implemented and interpreted to form a conclusion as to the trauma type.  Along these lines, although differences were statistically significant, a discussion on potential rate of error (including false positive and false negative errors) would be useful.

Author Response

We would like to thank the reviewers for their valuable comments that enhanced the quality of our manuscript. We hereby outline the proposed changes for the editors' and referees' approval.

Paragraph inserted concerning the limitations, potential error rate, and future research as well as a paragraph on how the findings of this study may be used for forensic cases (all highlighted in yellow within the text).